# Surgical Instrument Detection Algorithm Based on Improved YOLOv7x

**DOI:** 10.3390/s23115037

**Published:** 2023-05-24

**Authors:** Boping Ran, Bo Huang, Shunpan Liang, Yulei Hou

**Affiliations:** 1School of Information Science and Engineering, Yanshan University, Qinhuangdao 066000, China; ranboping@stumail.ysu.edu.cn (B.R.); huangbo@stumail.ysu.edu.cn (B.H.); 2School of Mechanical Engineering, Yanshan University, Qinhuangdao 066000, China; ylhou@ysu.edu.cn

**Keywords:** deep learning, YOLOV7x, surgical instrument detection, computer vision

## Abstract

The counting of surgical instruments is an important task to ensure surgical safety and patient health. However, due to the uncertainty of manual operations, there is a risk of missing or miscounting instruments. Applying computer vision technology to the instrument counting process can not only improve efficiency, but also reduce medical disputes and promote the development of medical informatization. However, during the counting process, surgical instruments may be densely arranged or obstruct each other, and they may be affected by different lighting environments, all of which can affect the accuracy of instrument recognition. In addition, similar instruments may have only minor differences in appearance and shape, which increases the difficulty of identification. To address these issues, this paper improves the YOLOv7x object detection algorithm and applies it to the surgical instrument detection task. First, the RepLK Block module is introduced into the YOLOv7x backbone network, which can increase the effective receptive field and guide the network to learn more shape features. Second, the ODConv structure is introduced into the neck module of the network, which can significantly enhance the feature extraction ability of the basic convolution operation of the CNN and capture more rich contextual information. At the same time, we created the OSI26 data set, which contains 452 images and 26 surgical instruments, for model training and evaluation. The experimental results show that our improved algorithm exhibits higher accuracy and robustness in surgical instrument detection tasks, with F1, AP, AP50, and AP75 reaching 94.7%, 91.5%, 99.1%, and 98.2%, respectively, which are 4.6%, 3.1%, 3.6%, and 3.9% higher than the baseline. Compared to other mainstream object detection algorithms, our method has significant advantages. These results demonstrate that our method can more accurately identify surgical instruments, thereby improving surgical safety and patient health.

## 1. Introduction

In recent years, with the development of computer vision (CV), it has been widely used in the medical field, covering multiple aspects, such as medical image analysis [1,2], pathological diagnosis [3,4], medical image generation [5], and medical robots [6]. For example, Kamnitsas et al. [7] successfully achieved accurate segmentation of brain lesions by using a three-dimensional convolutional neural network and a full connection conditional random field; Zhang et al. [8] proposed a new attention residual network model (ARL-CNN), which successfully improved the accuracy of classification of skin diseases; Hooshangnejad et al. [9] proposed a CT synthesis method, based on deep learning (deepPERFECT), which can quickly generate the CT images needed in the radiotherapy process of pancreatic cancer for treatment planning and evaluation; Sharghi et al. [10] proposed an automatic surgical activity recognition system, based on deep learning, for surgical activity analysis and automatic recognition in robotic assisted surgery. CV not only can improve the quality and accuracy of medical images, help doctors diagnose diseases and formulate treatment plans more accurately, reduce the workload of doctors, and reduce the occurrence of medical accidents, but it can also provide technical support for medical robots, intelligent medical devices, and improve the level and efficiency of medical services, thereby promoting the intelligent process of medical treatment.

Surgical instruments play a vital role in surgeries, assisting doctors in performing surgical procedures, controlling bleeding, and reducing the duration of surgeries. They are essential tools for doctors to complete surgeries. The proper selection and use of surgical instruments directly affect the success rate of surgery and the patient’s recovery. Each type of surgical instrument has a specific function, and any omission or confusion of surgical instruments during surgery may result in surgical failure and unnecessary harm to the patient. Moreover, if surgical instruments are inadvertently left in the patient’s body after surgery, it may lead to consequences, such as pain, infection, tissue inflammation, organ damage, and even death, resulting in serious medical accidents. A study [11] in 2019 reported 308 incidents of medical accidents involving retained foreign objects in patients’ bodies during surgery from 2012 to 2018. Most of these incidents occurred in the operating room and led to five patient deaths, with failure to count surgical instruments being the primary cause of these events. Therefore, it is necessary to inspect and count surgical instruments before and after surgery to ensure the smooth progress of surgery and the safety of patients.

The traditional method for counting surgical instruments typically involves medical staff checking the name, quantity, and model of each instrument on a list to ensure consistency with the surgical plan, as well as inspecting them for damage. However, due to the wide variety of surgical instruments and the potential for confusion between instruments with similar appearances, as well as factors, such as patient cases, individuals, and environments, the efficiency and accuracy of manual instrument counting are limited [12]. This method also requires a high demand for human resources and is prone to errors and misjudgments, making it difficult to meet the needs of last counting. Although methods, such as optimizing instrument placement [13], improving inventory systems [14], and introducing equipment maps [15], have been proposed, the shortcoming of manual counting cannot be resolved. To improve the method of counting surgical instruments, ultra-high frequency radio frequency identification (RFID) technology has been introduced [16]. RFID uses radio waves to identify specific targets and read/write related data. After installing RFID tags on surgical instruments, the RFID reader can recognize and locate them. This technology has advantages, such as high real-time performance and ease of operation, but some special instruments are difficult to tag, and the cost of equipment and tags is relatively high. With the development of artificial intelligence, it is now possible to classify and count surgical instruments using computer vision technology. Lee et al. [17] conducted experiments on a surgical instrument recognition dataset, comparing and analyzing the performance differences of three common object detection algorithms, Faster R-CNN, Mask R-CNN, and SSD, and they discussed their applicability in different scenarios. Object detection algorithms, based on deep learning, can help medical staff count surgical instruments quickly and accurately by recognizing and processing instrument images, thereby improving the efficiency of the operating room and reducing the occurrence of medical accidents.

Although there has been some progress in visual recognition for surgical instrument counting, there are still some problems. In real scenarios, surgical instruments may be densely arranged or obstructed by each other, and they may be affected by different lighting environments, all of which can affect recognition accuracy. In addition, some surgical instruments have very similar appearances, making it difficult to distinguish them through visual recognition, so achieving accurate recognition in different surgical scenes remains a challenge. To address these issues, this paper improves the YOLOv7x [18] and applies it to surgical instrument detection tasks. First, we improve the backbone network of YOLOv7x by introducing the RepLK Block module to guide the network to learn more shape features, thus obtaining more detailed surgical instrument feature information. Second, we improve the neck network of YOLOv7x by introducing the ODConv convolution structure to enhance the model’s feature extraction and generalization abilities. Finally, we collected 452 images, containing 26 types of surgical instruments, and generated the OSI26 data set. Based on this data set, we conducted experimental analysis to evaluate the model’s performance and compared it with currently advanced object detection models.

The summary of contributions for this work are as follows:A surgical instrument detection model was proposed to help hospitals achieve automated surgical instrument counting.The model can accurately identify 26 commonly used surgical instruments and has good recognition performance for some instruments with similar shapes and dense occlusions.The OSI26 surgical instrument data set was created and made public (address: https://aistudio.baidu.com/aistudio/datasetdetail/198164, accessed on 4 April 2023). It contains 452 images, covering 26 types of surgical instruments, each appearing 60 times. To enhance the model’s recognition accuracy and generalization ability, the surgical instruments were placed, with varying levels of dense occlusion, and data were collected under multiple lighting conditions. In addition, various data augmentation techniques were used to expand the data set.RepLK Block and ODConv structures were introduced to further improve the recognition accuracy of the YOLOv7x model, and experiments were conducted on the OSI26 data set. The experimental results show that the improved model performs significantly better than the original model, and it also shows superior performance compared to other advanced object detection algorithms.

The remaining sections of this paper are organized as follows. In Section 2, we review the literature on surgical instrument detection and counting. Section 3 describes the process of data set creation and data augmentation and proposes a new surgical instrument detection algorithm. Section 4 analyzes the experimental results of the method, identifies the current limitations, and proposes future research directions. Section 5 summarizes our research achievements.

## 2. Related Work

Currently, deep learning-based computer vision technology has been widely used in the detection of surgical instruments. S. Wang et al. [19] proposed a deep learning-based multi-label classification method for detecting surgical instruments in laparoscopic surgical videos. This method combined VGGNet and GoogLeNet and integrated the results of the models through ensemble learning to obtain the final result. Y. Wang et al. [20] and Y. Zhou et al. [21], respectively, proposed real-time detection of surgical instruments based on the YOLOv4 and YOLOv5 models to assist surgeons in minimally invasive surgery. Kaidi Liu et al. [22] proposed an enhanced feature-fusion network (EFFNet) for real-time detection of surgical instruments during surgery, helping surgeons to obtain more comprehensive visual information. A. Jin et al. [23] proposed using region-based convolutional neural networks to track and analyze surgical instruments in surgical videos, and they automatically evaluated the performance of surgeons. Kurmann et al. [24] proposed a method for identifying and estimating the pose of surgical instruments in minimally invasive surgery scenarios. This method first detected surgical instruments based on RetinaNet and then estimated the three-dimensional pose of surgical instruments based on MASK R-CNN. These two network models used shared convolutional neural network layers, which can improve the accuracy of pose estimation while ensuring the accuracy of detection.

Accurate and reliable inventory of surgical instruments is an important task for ensuring surgical safety. However, manual counting requires a significant amount of time and effort. To address this issue, Wang et al. [25] proposed a template matching method, based on morphological skeleton extraction and pixel sliding retrieval, for the inventory of surgical instruments. Although this method is effective, its ability to handle complex scenes is limited, and there are still limitations in dealing with issues, such as occlusion and rotation. Moreover, the generation and updating of templates require manual operations, which consume time and manpower. Lu [26] improved the DIoU-NMS algorithm of YOLOv4 and adopted the *k*-means++clustering algorithm and adaptive gamma correction algorithm to improve the recognition accuracy of surgical instruments in densely arranged and severely reflective situations. However, this study only targeted seven surgical instruments and did not consider the identification of similar instruments. Zhang [27] proposed a surgical instrument recognition method based on fine-grained image classification. By introducing group-enhanced attention and adaptive selection kernel attention in the ResNet network module, and combining multi-scale feature learning, the model’s ability to learn and extract detailed features of surgical instruments is enhanced, effectively improving the accuracy of surgical instrument classification. This method has a good auxiliary support effect on the inventory and recognition classification of surgical instruments, but it does not take into account the situation where multiple surgical instruments are obstructed and tightly arranged. Liang [28] studied 10 types of surgical instruments and introduced the cavity space convolutional pooling pyramid structure (ASPP) and the attention mechanism module (CBAM) in YOLOv4 to improve the recognition accuracy of surgical instruments. They achieved accurate classification of surgical scissors by optimizing the efficiency network, thereby solving the problem of identifying similar instruments under occlusion. However, the number of surgical instruments studied was relatively small, and the recognition performance of surgical instruments under densely arranged and different lighting conditions was not explored.

Based on the aforementioned issues, we have improved the YOLOv7x model and created a data set, containing 26 surgical instruments. We investigated the model’s ability to recognize surgical instruments with similar shapes and explored the effectiveness of instrument recognition under different lighting and dense occlusion conditions.

## 3. Materials and Methods

### 3.1. Image Acquisition

To conduct our research, we selected 26 representative surgical instruments from the basic orthopedic surgical instrument set, including surgical scissors, forceps, hook, bone knives, bone forceps, periosteal elevator, nerve stripper, surgical clamps, surgical knife handle, aspirator tip, scrapers, and spreaders, among other types. To facilitate description, we assigned them alphabetical labels from a to z, as shown in Figure 1. It is worth noting that we deliberately chose multiple instruments with very similar shapes to study the issue of identifying similar instruments. For instance, (a) and (b) belong to surgical scissors, (l) and (o) belong to bone knives, and (q), (r), and (s) belong to periosteal elevator, which only differ slightly in their head shapes. (h) and (p) are both bent vascular forceps, differing only in size, while (x) and (y) both belong to scalpel handles and have very similar external shapes. This makes them easily confused during actual counting and requires careful scrutiny by professionals.

We used different arrangements and combinations to capture images of the 26 types of surgical instruments mentioned above, including images of single instruments placed at different angles, multiple instruments with crossing occlusion but not arranged densely, multiple instruments with crossing occlusion and arranged densely, multiple instruments without crossing occlusion but arranged densely, and multiple instruments without crossing occlusion and not arranged densely. In addition, to increase the diversity of data, we captured instrument images in different lighting environments and collected images of both sides of the instruments, as shown in Figure 2. To ensure balanced data, we captured 60 images for each type of instrument, resulting in a total of 452 images. Of these, there were 156 images of single instruments, 72 images of multiple instruments without crossing occlusion, and 224 images of multiple instruments with crossing occlusion.

To ensure consistency in the data set, the pixel size of each image was set to 1980 pixels × 1080 pixels. The images were manually annotated using LabelImg software and saved in XML format. However, due to the small number of images in the data set, the model’s generalization ability and detection accuracy may be weak. Therefore, moderate data augmentation methods were adopted to improve the performance of the model. Compared to traditional data augmentation methods, online data augmentation has higher flexibility and real-time performance, and it can be performed on-demand without preprocessing [29]. As shown in Figure 3, basic data augmentation operations were performed on the original image data, including translation, scaling, flipping, brightness, hue, and saturation transformations, and corresponding probabilities were set. Mixup [30] and Mosaic [31] data augmentation methods were also introduced and applied with a certain probability to enhance the diversity of the data set and the generalization ability of the model. By using online data augmentation, the data set can be expanded to nearly five times its original size. The original data was divided into training, validation, and testing sets in a ratio, close to 8:1:1, while trying to maintain the balance of different category data, so that the distribution in the training, validation, and testing sets is roughly the same.

### 3.2. Construction of Surgical Instrument Detection Model

#### 3.2.1. YOLOv7x

YOLOv7 exhibits superior performance in both inference speed and accuracy in object detection tasks through strategies, such as Effective Layer Aggregated Network (E-ELAN), model scaling, reparameterization, auxiliary training heads, and multi-label assignment [18]. The YOLOv7 series models are arranged from small to large in terms of network depth and width, which are YOLOv7-tiny, YOLOv7, YOLOv7x, YOLOv7-W6, YOLOv7-E6, YOLOv7-D6, and YOLOv7-E6E. In this paper, we chose the YOLOv7x model, which has high accuracy and fast detection speed, meeting the requirements of surgical instrument detection scenes. The model structure of YOLOv7x is shown in Figure 4.

The entire model consists of four parts: the input end (Input), the backbone network (Backbone), the neck (Neck), and the prediction head (Head) [32]. The image input to the model is scaled to a uniform pixel size and enters the backbone network. The backbone network is mainly composed of CBS convolutional layers, E-ELAN modules, MPConv modules, and SPPCSPC modules. Among them, CBS convolutional layers are used to extract image features preliminarily; E-ELAN is improved on the basis of ELAN, using similar feature aggregation and feature transfer processes, controlling different lengths of connection paths, thereby obtaining richer features, and improving the calculation and utilization efficiency of model parameters; MPConv extracts features through two branches up and down, using both max pooling and convolution operations, and it fuses features through the Concat operation to enable the network to extract more effective information; the SPPCSPC module fuses feature information of different receptive fields, avoiding redundant feature extraction from the image and improving the expressive ability of feature maps.

The model’s neck uses the Path Aggregation Feature Pyramid Network (PAFPN) [33] to extract semantic features through a top-down path and combine them with precise localization information. It also enhances the bottom-up path to shorten the information path between low-level and top-level features, achieving efficient fusion of features at different levels. The prediction head adopts the RepConv structure without identity connections, which uses reparameterization to obtain more information and higher accuracy and predicts on three different-sized feature maps.

#### 3.2.2. RepLKNet

RepLKNet [34] uses large-size convolutional kernels to better preserve structural information in images while reducing information loss and blurring. The use of repeated layer structures further increases the depth and complexity of the network, improving the performance of image classification. Introducing large convolutional kernels significantly improves the effective receptive field compared to stacking small convolutions, guiding the network to learn more shape features. The network structure of RepLKNet is shown in Figure 5.

The network architecture of RepLKNet mainly consists of the RepLK Block, ConvFFN, and Transition Block. The RepLK Block contains deep convolution and shortcut connections. The use of large convolutional kernels may result in difficulty in capturing local features. The shortcut connections explicitly make the model a combination of models with different receptive field sizes, allowing for improvement in larger receptive fields without losing the ability to capture small-scale features. RepLKNet employs multiple repeated convolutional layers, each of which includes a large-sized convolutional kernel, a batch normalization layer, and a ReLU activation function. This repeated layer structure can make the network deeper, while the large-sized convolutional kernel can extract more abundant features, thereby improving the performance of image classification.

#### 3.2.3. ODConv

ODConv [35] utilizes a novel multi-dimensional attention mechanism and parallel strategy, considering four dimensions, including the number of convolutional kernels, spatial size, input channel, and output channel, so as to learn complementary attention and achieve dynamic convolution in all dimensions, thereby improving the flexibility and accuracy of the model. Unlike traditional convolutional neural networks that perform convolution operations with fixed convolutional kernels, ODConv makes each convolutional kernel a learnable parameter and can use different convolutional kernels at different positions to capture local features of the image and improve the accuracy of image recognition and object detection. it can also handle multi-channel inputs, making each channel have independent convolutional kernels to improve the diversity of features and the generalization ability of the model. As a direct substitute for traditional convolution, it can be embedded in most CNN architectures, as shown in Figure 6.

It can be described in the following form:(1)$$y=(\alpha_{s1}⊙\alpha_{c1}⊙\alpha_{f1}⊙\alpha_{\omega 1}⊙W_{1}+\ldots +\alpha_{sn}⊙\alpha_{cn}⊙\alpha_{fn}⊙\alpha_{\omega n}⊙W_{n})*x$$
$\alpha_{si},\;\alpha_{ci},and\;\alpha_{fi}$, respectively, represent attention along the spatial, input channel, and output channel dimensions, and $\alpha_{\omega i}$ represents the attention scalar of the convolutional kernel $W_{i}$. These four types of attentions complement each other and are sequentially accumulated onto the convolution kernel Wi, based on position, channel, filter, and kernel order, making the convolution operation different for all spatial positions, input channels, filters, and all kernels of input x. Therefore, ODConv can significantly enhance the feature extraction capability of traditional CNN convolution operations, capturing richer contextual information. Moreover, by using fewer convolutional kernels, ODConv can achieve comparable or even better performance than DyConv [36] and CondConv [37].

### 3.3. Improved Model

The improved structure of the YOLOv7x algorithm, as shown in Figure 7, enables real-time detection of multiple surgical instruments, suitable for the scenario of surgical instrument counting. We introduced the RepLK Block module into the backbone network of the original YOLOv7x, effectively improving the network’s perception range by utilizing the advantages of large convolution kernels, thereby obtaining more shape features and detailed information of surgical instruments. By fusing the feature information extracted by this module with the feature information extracted by the original network path, we can effectively identify surgical instruments with similar shapes and improve the recognition accuracy of surgical instruments in densely occluded situations. In order to further optimize the model and enhance the algorithm’s robustness, we introduced the full-dimensional dynamic convolution structure (ODConv) into the neck network of YOLOv7x. This structure can obtain feature information of different dimensions and enhance the model’s feature extraction and generalization capabilities using dynamically changing convolution kernel structures, thereby improving the recognition accuracy of surgical instruments under different lighting conditions and better meeting the demands of real-world scenarios.

## 4. Results and Discussion

The experimental environment is built on the Windows operating system, Python 3.8.0, PyTorch 1.11.0 deep learning framework, and Cuda 11.3. The GPU model is a RTX 3090 (with 24 GB of memory), and the CPU model is an Intel (R) Xeon (R) Platinum 8255C CPU @ 2.50 GHz. The model is trained with an initial learning rate (lr0) of 0.01, using a cosine annealing learning rate decay strategy with a cosine annealing hyperparameter (lrf) of 0.1. The loss function is optimized using the stochastic gradient descent (SGD) method, with a weight decay coefficient of 0.0005 and a learning rate momentum parameter of 0.937. Warm-up learning is performed in the first three epochs, with a warm-up learning rate of 0.8 and a warm-up learning rate momentum of 0.8. The input image size is set to 640 × 640, and the batch size for training is 16. A total of 150 epochs are trained, using transfer learning [38] to pre-train the network by loading the YOLOv7x.pt weight file to accelerate learning.

### 4.1. Evaluation Indicators

The experiment uses precision P, recall R, F1 (F-score), and mean Average Precision mAP as indicators to evaluate the performance of surgical instrument detection. Precision reflects the proportion of true positive samples in the predicted positive samples, while Recall reflects the proportion of correctly predicted positive samples in the total positive samples [39]. TP (True Positive) represents the number of samples correctly classified as positive, FP (False Positive) represents the number of samples incorrectly classified as positive, TN (True Negative) represents the number of samples correctly classified as negative, and FN (False Negative) represents the number of samples incorrectly classified as negative. The formulas for calculating precision P, recall R, and F1 are as follows.
(2)$$P(\%)=\frac{TP}{TP+FP}\times 100$$
(3)$$R(\%)=\frac{TP}{TP+FN}\times 100$$
(4)$$F1(\%)=\frac{2PR}{P+R}$$

mAP (mean average precision) is an important indicator for measuring detection accuracy in object detection, which represents the average AP (average precision) of each category. The AP value of each category can be calculated by integrating the PR (Precision-Recall) curve, and the calculation formula is as follows.
(5)$$AP=\int_{0}^{1}PdR$$
(6)$$mAP=\frac{1}{n}\sum_{i=1}^{n}AP_{i}$$

In the above formula, n represents the number of categories. We utilize common object detection evaluation metrics from the COCO evaluation, including AP, AP50, AP75, APS, APM, and APL. AP represents the mean average precision (mAP) across all categories. AP50 and AP75 represent the AP values at IoU thresholds of 0.5 and 0.75, respectively. APS represents the AP value for object bounding boxes with a pixel area smaller than $32^{2}$. APM represents the AP value for object bounding boxes with a pixel area ranging from $32^{2}$ to $96^{2}$. APL represents the AP value for object bounding boxes with a pixel area larger than $96^{2}$. Since the surgical instruments captured in our images are all large objects with bounding boxes exceeding $96^{2}$ in pixel area, it is not meaningful to compute APS and APM metrics. Additionally, for our data set, the APL value is essentially consistent with the AP value.

### 4.2. Ablation Experiments

To validate the effectiveness of the proposed improvement methods for surgical instrument category detection, we conducted ablation experiments. In these experiments, we tested each improvement method individually. “√” indicates that the corresponding improvement method was added, while “-” indicates that the method was not used. The experiments were conducted on the OSI2 surgical instrument data set, with input images uniformly resized to 640 × 640 and a batch size of 16. We utilized the pre-trained model YOLOv7x and trained the models for a total of 150 epochs. The performance comparison of different improvement methods on the validation set is presented in Table 1.

Based on the data in Table 1, it can be seen that the various improvements made to the model have achieved good results. By adding only the full-dimensional dynamic convolution structure to the original model, the model can obtain more abundant feature information, and the detection performance is optimized. Compared with the original model, the F1, AP, AP50, AP75, and APL are improved by 3.9%, 2.7%, 3.2%, 2.3%, and 2.7%, respectively. By adding only the RepLK Block to the original model, the model’s detection performance is significantly improved. Compared with the original model, the F1, AP, AP50, AP75, and APL are improved by 5.9%, 3.1%, 3.7%, 3.7%, and 3.1%, respectively. By integrating all the modules, the model’s detection performance is further improved, and compared with the original model, the F1, AP, AP50, AP75, and APL are improved by 7.5%, 4.8%, 5.2%, 5.3%, and 4.8% respectively, with a more significant improvement effect. In addition, during the model training process, the changes in the AP and AP50 of different improved models in the validation set are shown in Figure 8.

As shown in Figure 8, the improved model demonstrates higher accuracy and faster convergence speed compared to the original model, resulting in a significant enhancement in model performance. To further validate the effectiveness of the improvement methods, we conducted tests on the test set using different improvement approaches, and the results are organized in Table 2.

According to the data in Table 2, it can be observed that the improved model exhibits good performance on the test set. Compared to the original model, adding the full-dimensional dynamic convolution structure improves F1, AP, AP50, AP75, and APL by 3.1%, 2.6%, 2.8%, 3.4%, and 2.6%, respectively. Adding the RepLK Block improves F1, AP, AP50, AP75, and APL by 4.8%, 2.6%, 2.7%, 3.1%, and 2.6% respectively. What is more, combining these two improvement methods results in a performance increase of 4.6% in F1, 3.1% in AP, 3.6% in AP50, 3.9% in AP75 and 3.1% in APL.

To compare the improvement effects of the model more intuitively, we selected densely unoccluded, sparsely occluded, and densely occluded surgical instrument images from the test set for testing. The confidence threshold of the model was set to 0.25, and the IOU threshold was set to 0.45. The results are shown in Figure 9. The yellow boxes represent overlapping detection anchor boxes, red arrows indicate recognition errors, orange arrows indicate instruments not detected, “Predicted” represents the predicted category, and “True” represents the true category. For convenience, the instrument categories were labeled from 1 to 26.

In Figure 9, the first column displays the detection results of the YOLOv7x model, while the second column shows the detection results of the improved model. It can be observed that the original model tends to miss detections when surgical instruments are occluded. There are also several instances of overlapping anchor boxes, and the model exhibits weaker recognition capability for similar instruments. For example, instrument 4 is mistakenly identified as instrument 8, instrument 8 is mistakenly identified as instrument 3, and instrument 19 is mistakenly identified as instrument 17. In contrast, the improved model shows a noticeable enhancement in the recognition of surgical instruments. It does not miss any detections, with only one misclassification, and it significantly reduces the occurrence of overlapping anchor boxes. Therefore, the improved model exhibits a clear advantage in the task of surgical instrument recognition.

### 4.3. Model Performance Comparison

In order to comprehensively demonstrate the advantages of the improved algorithm, comparative experiments were conducted with classical object detection algorithms, such as YOLOv5l, YOLOv7, YOLOX-tiny, YOLOv6n, Faster RCNN, DETR, and Dynamic RCNN. Transfer learning was performed on the official pre-trained models, using the same data set, training epochs, and input image sizes. The performance of different models on the validation set is shown in Table 3.

The results in Table 3 indicate that the improved model, after training, achieves the best performance on the validation set, with AP, AP50, AP75, and APL reaching 92.0%, 99.8%, 98.3%, and 92.0%, respectively. Compared to YOLOv7, the improved model showed improvements of 9.3%, 9.1%, 10.8%, and 9.3% in terms of AP, AP50, AP75, and APL, respectively. Compared to YOLOv5l, the improved model demonstrated improvements of 0.5%, 0.9%, 1.2%, and 0.5% in AP, AP50, AP75, and APL, respectively. The performance of different models on the test set is shown in Table 4.

From the data in Table 4, it can be observed that the improved model also performs the best on the test set, with AP, AP50, AP75, and APL reaching 91.5%, 99.1%, 98.2%, and 91.5%, respectively. In terms of AP and APL metrics, the improved model shows a 10% improvement over Dynamic RCNN and a 0.8% improvement over YOLOv5l. Regarding the AP50 metric, the improved model exhibits an 8.6% improvement over YOLOv7 and a 0.6% improvement over Faster RCNN. For the AP75 metric, the improved model demonstrates an 8.8% improvement over YOLOv7 and a 0.7% improvement over Faster RCNN. Although Faster RCNN also achieves good detection results, it is a two-stage object detection algorithm with a larger model size, resulting in slower detection speed. Therefore, it is not suitable for real-time detection applications. In contrast, the model proposed in this paper is a one-stage object detection algorithm with a faster detection speed.

In hospitals, the commonly used method for surgical instrument counting is the double-checking method, which requires two surgical personnel to simultaneously count and cross-check the instruments to ensure accuracy. Typically, it takes 1–2 s for one person to manually count a surgical instrument. In contrast, the improved model proposed in this study can effectively identify six to seven images per second with a high recognition accuracy. Each image contains three to ten surgical instruments. Therefore, it can significantly assist in the manual counting of surgical instruments, greatly improving work efficiency, as well as reducing counting errors, thereby ensuring surgical safety.

### 4.4. Confusion Matrix Evaluation

We also evaluated the classification performance of the improved model using the confusion matrix for surgical instrument classification, as shown in Figure 10. In Figure 10, the left image represents the classification results of the original YOLOv7x model, while the right image represents the classification results of the improved model.

From Figure 10, It can be observed that the proposed improved model shows a significant improvement in the classification performance of surgical instruments with the labels 1, 2, 3, 4, 8, 16, 17, and 19, thereby enhancing the model’s recognition effectiveness. However, the improved model still requires further improvement in the classification performance for surgical instruments with the labels 7, 12, 17, 18, and 19.

### 4.5. Limitation and Discussion

This paper proposes an improved model based on Yolov7x for detecting surgical instruments during the instrument counting phase, aiming to improve counting efficiency and reduce error probability. Although the current improved model has achieved good results, it still has certain limitations.

Firstly, the improved model exhibits high accuracy in recognizing the 26 surgical instruments studied in this research. However, there may still be recognition errors for some instruments that have very similar appearances, especially for the (q) and (s) instruments depicted in Figure 1, which are bone dissector instruments with subtle differences (instruments numbered 17 and 19). As shown in Figure 11, the instrument labeled as 19 is mistakenly identified as the instrument labeled as 17 in both images.

Secondly, the proposed method in this paper is only applicable to common surgical instruments used in operating rooms. For special types of surgical instruments, such as those used in neurosurgery or microsurgery, dedicated counting methods need to be developed. Future research can focus on these specific types of instruments to devise counting approaches that are tailored to their characteristics, thus enhancing surgical safety and accuracy.

Lastly, the method presented in this paper is implemented based on computer vision technology. Although efforts have been made to consider factors, such as lighting, angles, and camera resolution during image data collection, there may still be counting errors caused by lighting variations, viewing angles, or camera limitations. Therefore, future research can explore new image processing techniques and deep learning models to improve the accuracy and reliability of surgical instrument counting.

## 5. Conclusions

Based on the real-time and low false positive requirements for surgical instrument detection in actual application scenarios, this paper proposes an improved YOLOv7x algorithm based on RepLK Block and ODConv to address issues such as dense arrangement of surgical instruments, mutual occlusion, difficulty in distinguishing similar instruments, and varying lighting conditions. The improved algorithm introduces the RepLK Block module into the original backbone network, which utilizes the advantages of large convolutional kernel structure to reduce the loss and blurring of image feature information, enhance the effective receptive field of the model, and thus improve the detection performance of surgical instruments under dense occlusion and the recognition accuracy of similar instruments. At the same time, the neck network introduces the ODConv structure, a full-dimensional dynamic convolution, to obtain more abundant feature information and enhance the robustness of the algorithm. The ODConv structure adjusts the convolutional kernels dynamically by obtaining multi-dimensional feature information, reducing the limitations of feature extraction ability in traditional convolutional neural networks, and improving the generalization ability of the model’s detection, thus further improving the accuracy of surgical instrument detection. After testing on the OSI26 data set, the improved algorithm achieved excellent performance in terms of precision, recall, F1 score, AP, AP50, AP75, and APL. The respective values are 92.6%, 97.0%, 94.7%, 91.5%, 99.1%, 98.2%, and 91.5%. These results represent improvements of 4.2%, 5.1%, 4.6%, 3.1%, 3.6%, 3.9%, and 3.1% compared to the baseline model. Compared with other object detection algorithms, such as YOLOv5l, YOLOv7, YOLOX-tiny, YOLOv6n, Faster RCNN, DETR, and Dynamic RCNN, the improved algorithm proposed in this paper also has significant advantages. The number of parameters in the improved model is 84.0MB, and it can detect six to seven densely arranged surgical instrument images per second, helping medical staff to quickly and accurately identify surgical instruments, improving the efficiency and accuracy of counting, and reducing the occurrence of medical accidents.

## Figures and Tables

**Figure 1 sensors-23-05037-f001:**
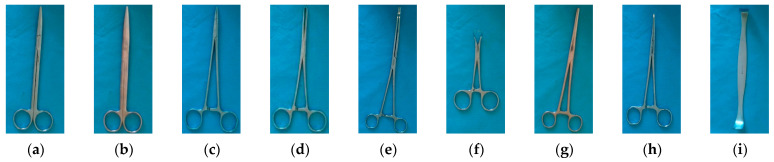
Pictures of different types of surgical instruments: (**a**) Comprehensive tissue scissors; (**b**) Stitch scissors; (**c**) Needle-holding pliers; (**d**) Tissue forceps; (**e**) Sponge pliers; (**f**) Towel clamp; (**g**) Buckle clamp; (**h**) Curved vessel forceps 1**#**; (**i**) Thyroid retractor; (**j**) Wire cutter; (**k**) Straight rongeur; (**l**) Bone knife 1**#**; (**m**) Double joint double angle handle rongeur; (**n**) Wound hook; (**o**) Bone knife 2**#**; (**p**) Curved vessel forceps 2**#**; (**q**) Periosteum elevator 1**#**; (**r**) Periosteum elevator 2**#**; (**s**) Periosteum elevator 3**#**; (**t**) Suction aspirator tip; (**u**) Nerve dissector; (**v**) Curette; (**w**) Smooth Forceps; (**x**) Surgical Knife Handle 1**#**; (**y**) Surgical Knife Handle 2**#**; (**z**) Guide wire.

**Figure 2 sensors-23-05037-f002:**
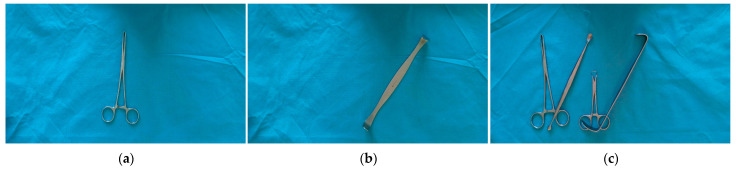
Collected images of surgical instruments: (**a**) Image of tissue forceps; (**b**) Image of thyroid retractor; (**c**) Image of multiple instruments that are arranged in a crossed and overlapping, but not densely packed, manner; (**d**) Image of multiple instruments that are arranged in a crossed and densely packed manner; (**e**) Image of multiple instruments that are arranged in a non-overlapping and densely packed manner; (**f**) Image of multiple instruments that are arranged in a non-overlapping and not densely packed manner.

**Figure 3 sensors-23-05037-f003:**
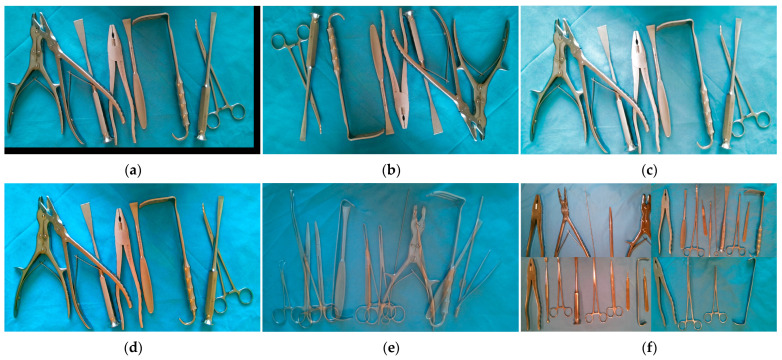
Data argumentation: (**a**) Translation; (**b**) Vertical flipping; (**c**) Brightness transformation; (**d**) Saturation transformation; (**e**) Mixup; (**f**) Mosaic.

**Figure 4 sensors-23-05037-f004:**
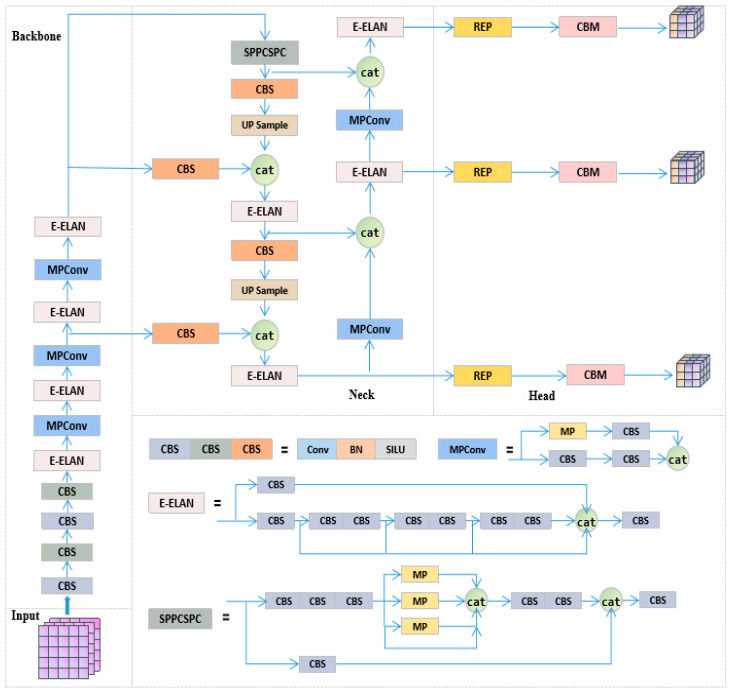
Structure of YOLOv7x.

**Figure 5 sensors-23-05037-f005:**
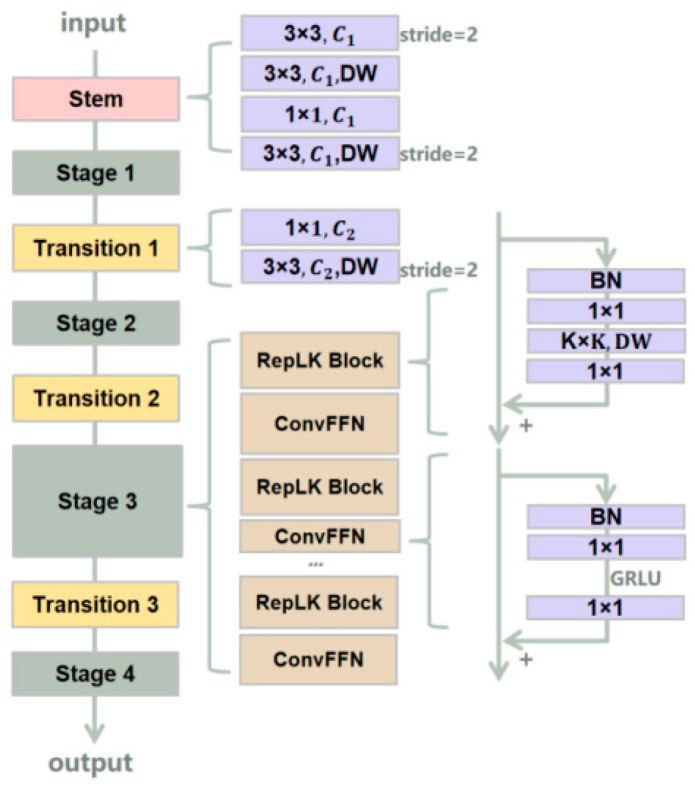
RepLKNet network structure.

**Figure 6 sensors-23-05037-f006:**
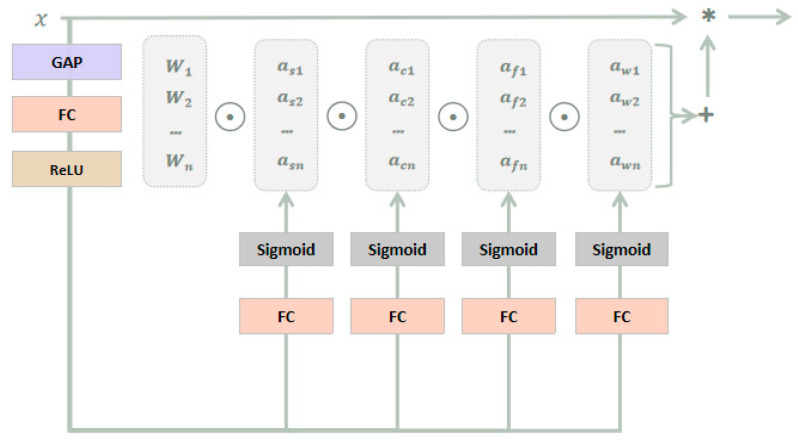
Schematic diagram of ODConv. The symbol “*” represents the convolution operation.

**Figure 7 sensors-23-05037-f007:**
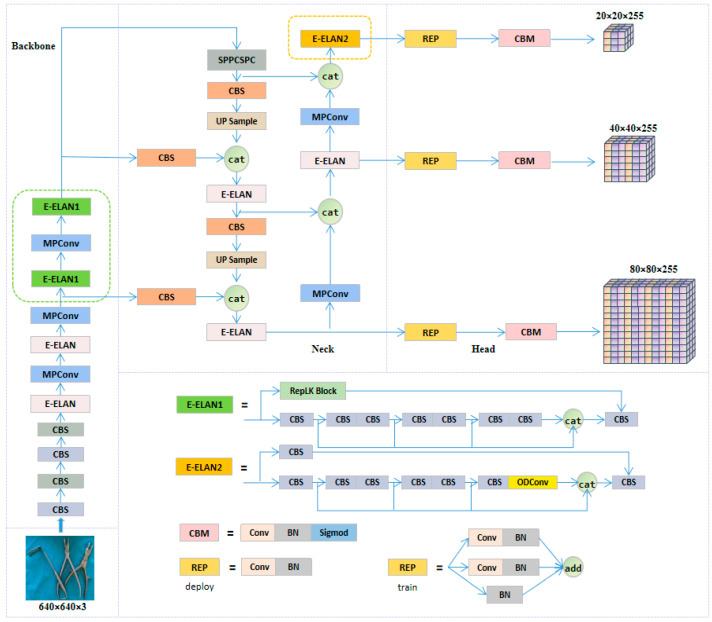
Structure of the improved YOLOv7X model.

**Figure 8 sensors-23-05037-f008:**
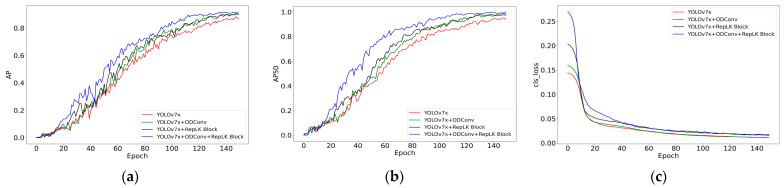
Comparison of different improvement methods on the validation set: (**a**) Comparison of the AP values for the different improved model; (**b**) Comparison of the AP50 values for the different improved model; (**c**) Comparison of the classification loss for the different improved model.

**Figure 9 sensors-23-05037-f009:**
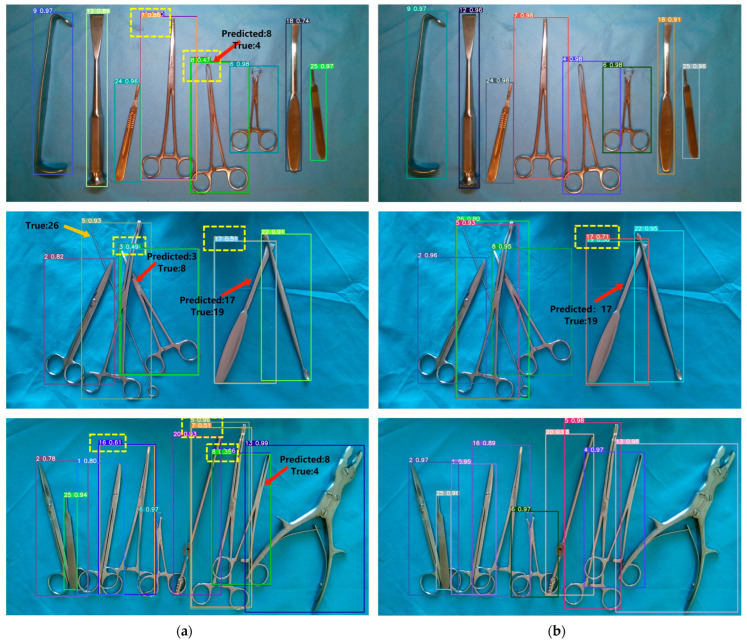
Comparison of the effects between the original model and the improved model: (**a**) Detection effect of YOLOv7x; (**b**) Detection effect of the improved model.

**Figure 10 sensors-23-05037-f010:**
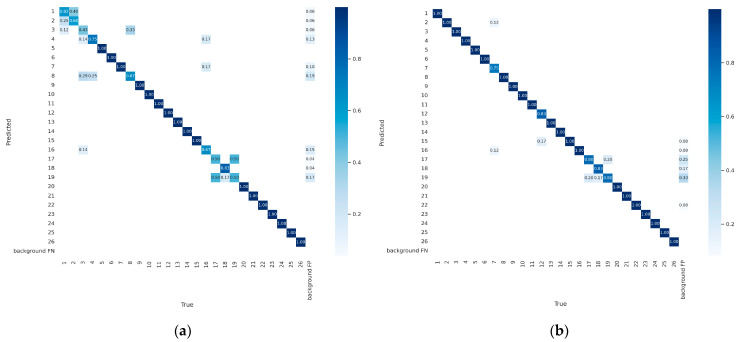
Confusion Matrix between the original model and the improved model: (**a**) Detection effect of YOLOv7x; (**b**) Detection effect of improved model.

**Figure 11 sensors-23-05037-f011:**
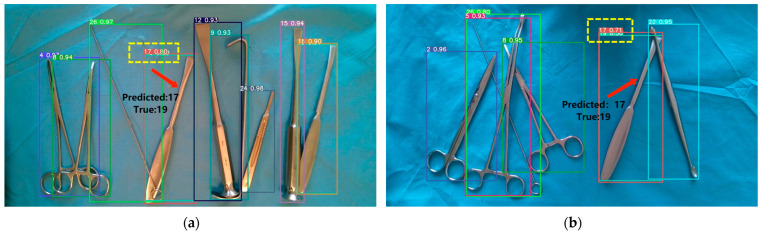
Images with incorrect surgical instrument recognition: (**a**) The instrument labeled as number 19 is mistakenly identified as instrument number 17 (1); (**b**) The instrument labeled as number 19 is mistakenly identified as instrument number 17 (2).

**Table 1 sensors-23-05037-t001:** Comparison of the performance of the different improvement methods on the validation set.

RepLK Block	ODConv	P	R	F1	AP	AP50	AP75	APL
-	-	83.9%	94.2%	88.8%	87.2%	94.6%	93.0%	87.2%
-	√	91.9%	93.5%	92.7%	89.9%	97.8%	95.3%	89.9%
√	-	91.9%	97.6%	94.7%	90.3%	98.3%	96.7%	90.3%
√	√	95.1%	97.6%	96.3%	92.0%	99.8%	98.3%	92.0%

**Table 2 sensors-23-05037-t002:** Comparison of the performance of the different improvement methods on the test set.

RepLK Block	ODConv	P	R	F1	AP	AP_50_	AP_75_	AP_L_
-	-	88.4%	91.9%	90.1%	88.4%	95.5%	94.3%	88.4%
-	√	89.4%	97.3%	93.2%	91.0%	98.3%	97.7%	91.0%
√	-	94.3%	95.4%	94.9%	91.0%	98.2%	97.4%	91.0%
√	√	92.6%	97.0%	94.7%	91.5%	99.1%	98.2%	91.5%

**Table 3 sensors-23-05037-t003:** Performance of different models on the validation set.

Models	Model Backbone	Image Size	AP	AP50	AP75	APL	Iteration Number
YOLOv5l	CSPDarknet53	640 × 640	91.5%	98.9%	97.1%	91.5%	150
YOLOv7	CSPDarknet53		82.7%	90.7%	87.5%	82.7%	
YOLOv7x	CSPDarknet53		87.2%	94.6%	93.0%	87.2%	
YOLOX-tiny	Darknet53		79.2%	96.6%	92.3%	79.2%	
YOLOv6n	EfficientRep		90.1%	98.7%	96.0%	90.1%	
Faster RCNN	ResNet101		87.3%	97.9%	96.6%	87.3%	
DETR	ResNet50		86.1%	94.8%	93.6%	86.1%	
Dynamic RCNN	ResNet50		86.4%	98.4%	96.0%	86.4%	
Improved YOLOv7x	CSPDarknet53		92.0%	99.8%	98.3%	92.0%	

**Table 4 sensors-23-05037-t004:** Performance of different models on the test set.

Models	Image Size	Params	FLOPs	AP	AP50	AP75	APL
YOLOv5l	640 × 640	46.2 M	108.1 G	90.7%	97.0%	96.1%	90.7%
YOLOv7		36.9 M	104.7 G	83.5%	90.5%	89.4%	83.5%
YOLOv7x		71.3 M	189.9 G	88.4%	95.5%	94.3%	88.4%
YOLOX-tiny		5.06 M	6.45 G	77.6%	96.0%	90.8%	77.6%
YOLOv6n		4.63 M	11.36 G	88.0%	97.9%	94.1%	88.0%
Faster RCNN		60.5 M	283.1 G	88.0%	98.5%	97.5%	88.0%
DETR		41.3 M	91.6 G	86.9%	95.8%	92.6%	86.9%
Dynamic RCNN		41.3 M	206.8 G	81.5%	95.6%	93.2%	81.5%
Improved YOLOv7x		80.4 M	204.0 G	91.5%	99.1%	98.2%	91.5%

## Data Availability

Not applicable.

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
