# Peer review of "Surgical Instrument Detection Algorithm Based on Improved YOLOv7x"

_sensors, 2023, doi:10.3390/s23115037_

Round 1

Reviewer 1 Report

Authors in this research used an existing model YOLOv7x for Surgical instrument detection.

My major concerns are 

what is main contribution of this research, I am unclear ? state your key contributions with justification.

What dataset used for experiments, please mention.

Only YOLOv7x  & RepLKNet  employed is it fusion research ? please explain in detail. If authors improved then explain what is need of this research.

The proposed model is blur for readers so explain in details.

Moderate corrections required.

Author Response

Dear Reviewer,

Thank you very much for your diligent review of our research and valuable feedback. We greatly appreciate the valuable suggestions you have provided. We have made revisions to the paper and highlighted the modifications in blue throughout the document. In this response, we will address your suggestions and questions in detail and express our heartfelt gratitude for your guidance.

Here are the answers to the questions you raised:

Point 1: what is main contribution of this research, I am unclear ? state your key contributions with justification.

Response 1: In surgical procedures, the counting of surgical instruments is crucial. However, the current process is primarily manual, which is inefficient and prone to errors. Therefore, we aim to improve this process using computer vision technology. We have employed the YOLOv7x object detection algorithm for surgical instrument detection to assist in the manual counting of instruments. However, in practical scenarios, surgical instruments are often densely arranged or occluded, and they can be affected by varying lighting conditions. Additionally, some instruments have minimal differences in appearance and shape, which increases the difficulty of algorithm recognition. To address these challenges, we have made improvements and optimizations to the YOLOv7x algorithm.We introduced the RepLK Block module and ODConv structure to enhance the algorithm's performance. The RepLK Block module expands the receptive field and guides the network to learn more shape features, while the ODConv structure enhances the feature extraction capability of convolution operations, capturing richer contextual information. With these improvements, we have enhanced the accuracy and robustness of the surgical instrument detection algorithm, enabling its application in the counting of surgical instruments. We encourage you to refer to the introduction and related work sections of our paper for a more comprehensive understanding of our research.

Point 2: What dataset used for experiments, please mention.

Response 2: We used our self-created dataset OSI26 for model training and evaluation in the experiments. This dataset consists of 452 images and 26 types of surgical instruments.We have provided a detailed introduction and explanation of the dataset in Section 3.1 of the paper.You can find the dataset at the following address:

https://aistudio.baidu.com/aistudio/datasetdetail/198164。

Point 3: Only YOLOv7x  & RepLKNet  employed is it fusion research ? please explain in detail. If authors improved then explain what is need of this research.

Response 3: In this study, we have made improvements and optimizations based on the YOLOv7x algorithm. We found that incorporating the RepLKNet Block and ODConv module into the YOLOv7x model significantly improves the accuracy and robustness of the model.The RepLK Block module expands the receptive field and guides the network to learn more shape features, while the ODConv structure enhances the feature extraction capability of convolution operations, capturing richer contextual information. We have described and explained YOLOv7x, RepLKNet Block, ODConv, and the improved model in Section 3.2 of the paper.

Point 4: The proposed model is blur for readers so explain in details.

Response 4: In Section 3.2 of the paper, we have provided a description and explanation of YOLOv7x, RepLKNet Block, ODConv, and the improved model. For the structural diagram of our proposed improved model, you can refer to Figure 6 on page 10. In this section, we have also provided explanations for the role of each module. To demonstrate the effectiveness of the model improvements, we have conducted an analysis and discussion of the experimental results in Section 4 of the paper. Specifically, on lines 470 to 477 on page 15, we have explained the practical value of the improved model.

Once again, we appreciate your guidance and suggestions. We have carefully considered the questions you raised and made appropriate revisions and improvements in the article, highlighting the changes in blue. If you have any further questions or suggestions, please feel free to let us know. We would be happy to engage in further discussion.

Reviewer 2 Report

The manuscript presents an improved YOLOv7x object detection algorithm that accurately detects and counts surgical instruments, 

even in challenging situations, which can improve surgical safety and patient health.

The article to be well written and well executed.  The authors have done a commendable job in designing and conducting their experiments, and their results are clearly presented and well-supported. The experimental design is appropriate and the data analysis is sound. The authors have also provided a clear and comprehensive discussion of their results, which is a strength of the paper. One suggestion for improvement that I have is that the study could benefit from testing on a larger dataset. This would help to confirm the robustness of the findings and increase the generalizability of the results. However, I understand this may not have been possible due to limitations in data availability. Check the figure position.

Author Response

Thank you very much for your recognition and evaluation of our research article. Your valuable feedback is highly appreciated.

We are delighted that you found our writing and experimental design to be excellent. We have put a lot of effort into designing and conducting the experiments, and it is gratifying to receive clear support for the presentation of our results. The experimental design was appropriate, and the data analysis was reliable. We have provided a clear and comprehensive discussion of the results, which is a highlight of this article.

Point 1:One suggestion for improvement that I have is that the study could benefit from testing on a larger dataset. This would help to confirm the robustness of the findings and increase the generalizability of the results. However, I understand this may not have been possible due to limitations in data availability.

Response 1: The suggested improvement is to test our research on a larger dataset. This would help confirm the robustness of our research findings and enhance the generalizability of the results. However, due to limitations in the availability of surgical instrument data, we are unable to conduct testing on a larger scale dataset.We will strive to build a larger-scale dataset in future research to further support and expand upon our research findings.

Point 2: Check the figure position.

Response 2: Thank you for bringing up the issue regarding the placement of the figures. We have thoroughly reviewed and ensured the accurate placement of the figures.

Once again, we appreciate your guidance and suggestions. We have carefully considered the questions you raised and made appropriate revisions and improvements in the article, highlighting the changes in blue. If you have any further questions or suggestions, please feel free to let us know. We would be happy to engage in further discussion.

Author Response

Dear Reviewer,

Thank you very much for your thorough review and valuable feedback on our research. We greatly appreciate your insightful suggestions. Based on your recommendations, we have made revisions to the paper and highlighted the modifications in blue throughout the document. In this response, we will provide detailed answers to your suggestions and questions, and we sincerely thank you for your guidance.Here are the answers to the questions you raised:

Point 1: The introduction is written well. Literature review also provides sufficient information about previous studies.

Response 1: Thank you very much for your positive feedback on the introduction section of our article. We have made efforts to ensure that the introduction provides a clear overview of the research background and significance, while also engaging the readers with the topic of the paper. In this revision, we have further supplemented the introduction section and highlighted the newly added content in blue, which can be found in pages 2 and 3 of the paper.

Point 2: However, it would be nice if authors can provide related work section and a table to deeply analyze the methods used in previous studies and the drawbacks and advantages of similar studies.

Response 2: Thank you very much for your valuable suggestions. In this revision, we have added a section on related work, where we summarize the improvements and limitations of previous research methods and similar approaches. This sets the stage for introducing the specific focus of our study. Detailed information can be found on pages 3 and 4 of the paper.We are committed to providing a comprehensive literature review to ensure coverage of relevant previous studies related to our research topic. This helps establish a solid theoretical foundation for our work.

Point 3: I think it is necessary to give some details of the proposed surgical instrument detection framework by adding figures in Section 2.

Response 3: Thank you very much for your valuable suggestions. In this revision, we have made improvements to Figure 6 on page 10 by modifying the representation of the inputs and outputs in the framework diagram. As our proposed improved model is based on YOLOv7x, some details of the modules in Figure 6 can be referenced from Figure 3 on page 8.

Point 4: Please add example figures for labelling and data augmentation process in Section 2.1.

Response 4: Thank you very much for your valuable suggestions. In this revision, we have added example images of data augmentation in Figure 3 on pages 6 to 7. These examples demonstrate the effects of six types of data augmentation: Translation, Vertical flipping, Brightness transformation, Saturation transformation, Mixup, and Mosaic.

Point 5: Please add visual representation of Figure 8.

Response 5: Thank you very much for your revision suggestions. In this revision, we have redrawn the images in Figure 8 to present the detection results before and after the model improvements more intuitively and clearly. We have annotated the images with yellow rectangular boxes, arrows, and text. Please refer to pages 13 to 14, lines 420 to 441, for detailed information.

Point 6: Results section is weak, so more and convincing results must be added to meet the standard of the journal.

Response 6: Thank you very much for your revision suggestions. In this revision, we have added supplementary experimental content and included additional metrics such as F1, AP, AP50, AP75, APL, etc. We conducted ablation experiments and comparative experiments with other mainstream algorithms. Finally, we analyzed the confusion matrix to fully demonstrate the effectiveness of our proposed improved model. Please refer to pages 11 to 16, lines 351 to 489, for detailed information.

Point 7: I think it is necessary to compare your work with other similar works using Microsoft COCO benchmarks such as F-measure and others. Thus, please add tables or figures to prove your work's advantages over other similar works.

Response 7: Thank you very much for your valuable suggestions. In this revision, we have added metrics such as F1, AP, AP50, AP75, and APL to enhance the experimental results. We compared the performance of our improved model with the metrics of similar studies to demonstrate the advantages of our work over other similar research. Please refer to pages 11 to 15, lines 351 to 477, for detailed information.

Point 8: In general, Section 3 is must be modified with more evaluations and comparisons with other popular methods.

Response 8: Thank you very much for your valuable suggestions. In this revision, we have added metrics such as F1, AP, AP50, AP75, and APL, and evaluated and compared our improved model with other popular object detection algorithms: YOLOv5l, YOLOv7, YOLOX-tiny, YOLOv6n, Faster RCNN, DETR, and Dynamic RCNN. Please refer to pages 14 to 15, lines 442 to 477, for detailed information.

Point 9: Please add a "Limitation and Discussion" section to give a limitation of the proposed method and future research gaps in this field.

Response 9: Thank you very much for your valuable suggestions. In this revision, we have added a "Limitation and Discussion" section on page 16, specifically from line 490 to 515 in the paper. In this section, we comprehensively discuss the limitations of the proposed method and explore potential gaps and future directions for research in the field.

Point 10: How does this method work with higher resolution images?

Response 10: Thank you for your question. Our method is also applicable to higher-resolution images. The model architecture includes multiple convolutional layers and a feature pyramid structure, which can handle features at different scales. Higher-resolution images can be effectively processed using these hierarchical layers and feature pyramids.

Point 11: How can we implement this work in practice?

Response 11: Thank you for your question. We can develop a surgical instrument counting system based on the model proposed in this research. The system includes a user-designed software application, cameras, a broadcasting device, and a surgical instrument counting platform. Medical staff can place the surgical instruments on the detection platform, and the cameras will capture images of the instruments. By using this model, the instruments can be detected, and the current category and quantity can be counted. The results can be broadcasted through the broadcasting device. Medical staff can review the statistics and perform a manual verification, saving time in surgical instrument counting, and improving efficiency and safety.

Point 12: Authors should also check the article for typo errors and English grammar.

Response 12: Thank you very much for your valuable suggestions. With the help of two professional English teachers, we have reviewed and edited the entire paper, correcting spelling errors and addressing grammar issues.

The two papers you provided have been very helpful to us, and we sincerely thank you for your guidance and suggestions. We have carefully considered the questions you raised and made appropriate modifications and improvements in the article. If you have any further questions or suggestions, please feel free to let us know. We are more than happy to engage in further discussion.

Round 2

Reviewer 1 Report

Accepted

Aceptable